# The connecting cilium inner scaffold provides a structural foundation that protects against retinal degeneration

Olivier Mercey[1], Corinne Kostic[2], Eloïse Bertiaux[1], Alexia Giroud[1], Yashar Sadian[3], David C. A. Gaboriau[4¤], Ciaran G. Morrison[4], Ning Chang[5], Yvan Arsenijevic[5], Paul Guichard[1]*, Virginie Hamel[1]*

1 Department of Molecular and Cellular Biology, University of Geneva, Geneva, Switzerland, 2 Group for Retinal Disorder Research, Department of Ophthalmology, University Lausanne, Jules-Gonin Eye Hospital, Fondation Asile des Aveugles, Lausanne, Switzerland, 3 CryoGenic Facility, University of Geneva, Geneva, Switzerland, 4 Centre for Chromosome Biology, National University of Ireland Galway, Galway, Ireland, 5 Unit of Retinal Degeneration and Regeneration, Department of Ophthalmology, University Lausanne, Jules-Gonin Eye Hospital, Fondation Asile des Aveugles, Lausanne, Switzerland

¤ Current address: Facility for Imaging by Light Microscopy, Faculty of Medicine, Imperial College London, London, United Kingdom

* paul.guichard@unige.ch (PG); virginie.hamel@unige.ch (VH)

**Data Availability Statement:** All data are available in the main text or the supplementary information.

**Funding:** This work is supported by the ERC StG 715289 (ACCENT), the Swiss National Foundation

## Abstract

Inherited retinal degeneration due to loss of photoreceptor cells is a leading cause of human blindness. These cells possess a photosensitive outer segment linked to the cell body through the connecting cilium (CC). While structural defects of the CC have been associated with retinal degeneration, its nanoscale molecular composition, assembly, and function are barely known. Here, using expansion microscopy and electron microscopy, we reveal the molecular architecture of the CC and demonstrate that microtubules are linked together by a CC inner scaffold containing POC5, CENTRIN, and FAM161A. Dissecting CC inner scaffold assembly during photoreceptor development in mouse revealed that it acts as a structural zipper, progressively bridging microtubule doublets and straightening the CC. Furthermore, we show that *Fam161a* disruption in mouse leads to specific CC inner scaffold loss and triggers microtubule doublet spreading, prior to outer segment collapse and photoreceptor degeneration, suggesting a molecular mechanism for a subtype of retinitis pigmentosa.

## Introduction

The retina is a thin tissue lining the back of the eyeball that contains photoreceptor cells, which convert light inputs into electrical signals, a process crucial for the detection of visual stimuli. These highly specialized ciliated cells are partitioned into 2 main regions, a photosensitive outer segment (OS) and a photoreceptor inner segment, which are connected via a thin bridging structure known as the connecting cilium (CC) with its underlying ciliary basal body. This structural linker, emanating from a mature centriole, is made of 9 microtubule doublets (MTDs) that extend distally to form the axonemal stalk, the basis of the OS onto which hundreds of stacked membrane discs that contain phototransduction proteins are positioned [1–

(SNSF) PP00P3_187198 attributed to P.G., the Pro Visu Foundation attributed to P.G, V.H. and C.K., the Fondation Asile des Aveugles (fonds RO1011) attributed to Y.A., European Commission SEC-2009-4.3-02 (project 242361) attributed to C.M., the EMBO fellowship (ALTF-284-2019) to E.B. and the Novartis Foundation for medical-biological Research (18B112) attributed to P.G. The funders had no role in study design, data collection and analysis, decision to publish, or preparation of the manuscript.

**Competing interests:** Authors declare no competing interests.

**Abbreviations:** CC, connecting cilium; EM, electron microscopy; FWHM, full width at half maximum; MS, monomer solution; MTD, microtubule doublet; OS, outer segment; PBS, phosphate-buffered saline; ROI, region of interest; RT, room temperature; siRNA, small interfering RNA; TZ, transition zone; WT, wild type.

3]. Mutations in the gene that encodes the microtubule-binding protein FAM161A, which localizes at the CC, have been associated with the human pathology retinitis pigmentosa 28 (RP28), a subtype of retinitis pigmentosa, the most prevalent human inherited retinal disease with an incidence of 1/4,000 worldwide [4–11]. Mouse models of *FAM161A*-associated RP28 have revealed structural defects in the CC, with spread MTDs and disturbed membrane disc organization that underlie photoreceptor loss [9,12]. Similarly, mutations in the genes for the CC-localized proteins POC5, CENTRIN, and POC1B have all been associated with retinal pathologies displaying photoreceptor degeneration [13–15].

Recently, we localized these 4 proteins at the level of centrioles and composing the so-called inner scaffold, a structure connecting neighboring microtubule blades [16]. Depletion of inner scaffold components leads to centriole architectural defects, hinting at a role for this structure in the structural cohesion of the entire organelle [16,17]. These 4 inner scaffold proteins also being present in the CC, we hypothesized that a similar inner scaffold structure might exist at the level of the CC to provide the structural cohesion of the MTDs, thus ensuring OS integrity.

## Results

### Molecular mapping of the mammalian photoreceptor connecting cilium

To test this hypothesis, we set out to reveal the molecular architecture of the CC using and optimizing Ultrastructure Expansion Microscopy (U-ExM) [18] on adult mouse retinal tissue (**S1 Fig**). To validate our approach, we analyzed at low magnification the overall organization of the tissue after expansion and demonstrated the preservation of the different cell layers of the retina, notably the photoreceptor cells (**Fig 1A and 1B**). Moreover, staining photoreceptor cells for α/β-tubulin revealed microtubule structures at higher magnification, thus allowing the visualization of both mother and daughter centrioles, the CC, and the extending OS axoneme, which forms an enlarged region immediately after the CC, that we named the bulge region (**Fig 1B** and **S1 Fig**). Additionally, we examined the localization of the centriolar inner scaffold component POC5. We consistently found it in the central core region of centrioles and also clearly decorating the entire CC region, as previously reported [15,16] (**Fig 1B**). From this result, we next determined the expansion factor resulting from this optimized protocol using photoreceptor centriole diameter as a ruler and an indicator of structural integrity. By comparing it to measurements of human U2OS centrioles, we found an expansion factor of 4.2 on average (**S1 Fig**), matching the previously published values and thus validating our approach [16,19]. For the rest of the study, all the measured data were corrected for the expansion factor.

We next investigated the precise localization of the inner scaffold components POC5, CENTRIN, and FAM161A in adult mouse photoreceptors, with the exception of POC1B owing to the lack of appropriate antibodies. We also inspected the distribution of CEP290, known to localize along the CC from super-resolution microscopy [20], and Lebercilin (LCA5), a proposed FAM161A interactor [10], mutations in which have been linked to Leber congenital amaurosis, a retinopathy causing severe visual deficiency from the first year after birth [21,22]. Consistent with our previous work, all 3 inner scaffold proteins were found in the central core region of centrioles, where the inner scaffold structure lies [16] (**Fig 1C–1E**). In addition to confirming POC5, CENTRIN, and FAM161A localization along the CC, we observed that these proteins line the inner part of the microtubule wall, although FAM161A antibody displayed a weaker and punctate signal, probably due to partial epitope conservation from mouse to human (**Fig 1C–1E**). We also noted a gap between protein signals in the centriole and the CC that we hypothesize correspond to the distal end of the centriole, devoid of the centriolar inner scaffold structure [16], suggesting that these 2 regions are independent (**Fig 1B**, white

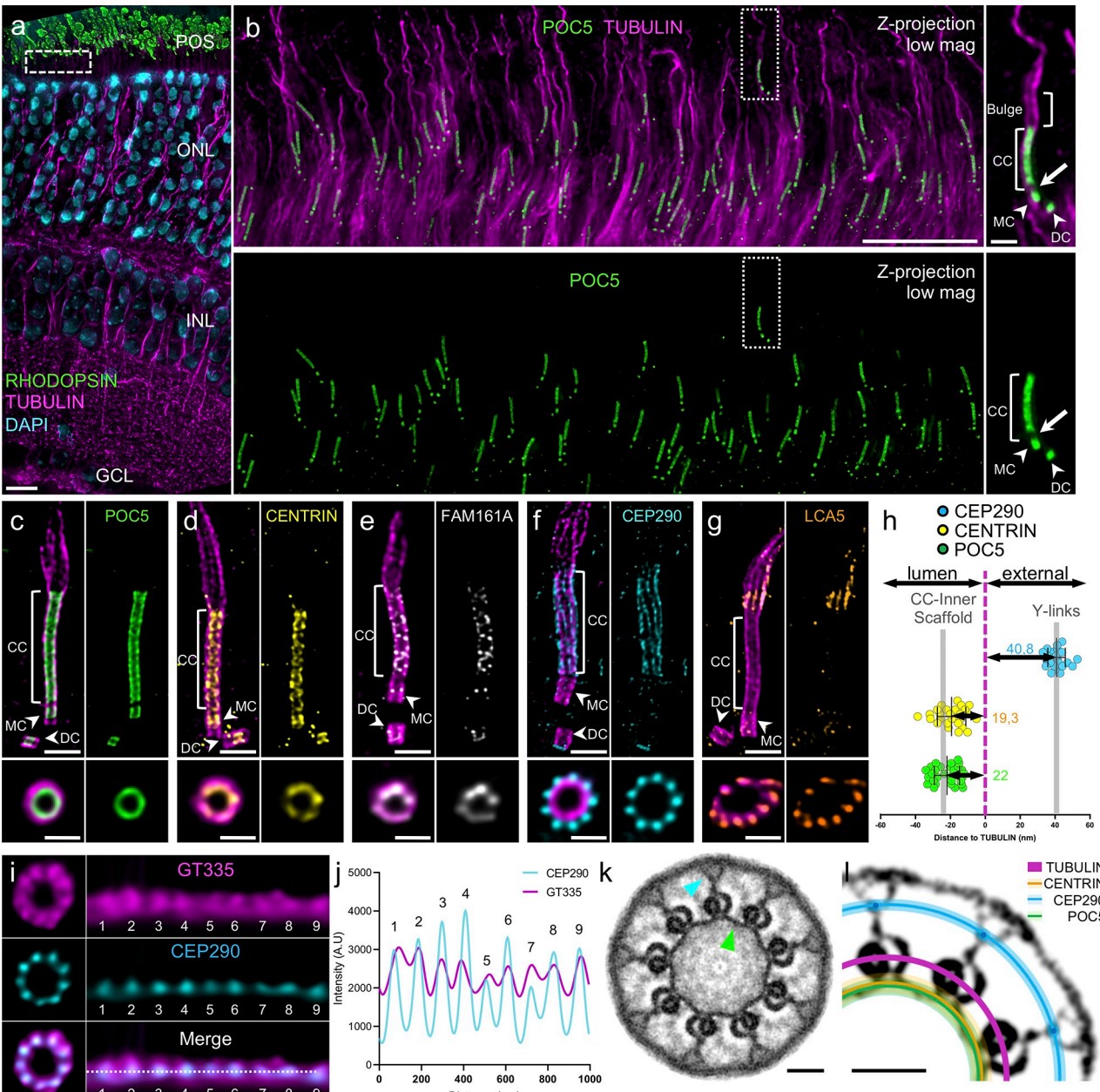

**Fig 1. Molecular mapping of the mammalian photoreceptor connecting cilium.** (**a**) Low magnification U-ExM image of a P14 mouse retina highlighting the preservation of the different retina layers. Note that the RPE layer is removed with the dissection. Scale bar: 10 μm. (**b**) Expanded P14 photoreceptor layer (equivalent region to the white dashed line depicted in (a)). Inset shows details of 1 photoreceptor cell. Arrowheads indicate centrioles. Arrow depicts the gap between centriole and CC POC5 signals. Scale bar: 5 μm (inset: 500 nm). (**c–g**) Confocal U-ExM images of adult photoreceptors stained for tubulin (magenta) and POC5 (c: green) or CENTRIN (d: yellow), FAM161A (e: gray), CEP290 (f: cyan), and LCA5 (g: orange). Lower panels show transversal views of the CC for each staining. Arrowheads indicate centrioles. Scale bars: side view = 500 nm; transversal view = 200 nm. (**h**) Distances between the maximum intensity of POC5 (green), CENTRIN (dark yellow), and CEP290 (cyan) compared to tubulin (magenta) calculated from transversal view images. Gray bars indicate the values obtained from the simulation in S4 Fig. $n \geq 2$ animals per staining. See S1 Table. (**i**) Transversal view (left) and polar transform (right) of GT335 (magenta) and CEP290 (cyan) signals revealing overlapping 9-fold symmetry. Scale bar: 200 nm. (**j**) Plot profiles of CEP290 (cyan) and GT335 (magenta) polar transform of (i). (**k**) Symmetrized EM image of a P14 CC transversal section revealing an inner ring decorating MTDs (green arrowhead) and Y-links bridging MTDs to the membrane (blue arrowhead). Scale bar: 50 nm. (**l**) Model representing relative positions calculated in (h) and (j) of POC5 (green line), CENTRIN (dark yellow line), CEP290 (cyan dot and line) to tubulin (magenta) on a contrasted symmetrized EM picture of a CC. Light color lines represent the SD for each protein. Scale bar: 50 nm. The data underlying all the graphs shown in the figure are included in the S1 Data file. CC, connecting cilium; DC, daughter centriole; GCL, ganglion cells layer; IPL, inner nuclear layer; MC, mother centriole; MTD, microtubule doublet; ONL, outer nuclear layer; POS, photoreceptor outer segment.

arrow). In contrast, we found that CEP290, capping the distal end of daughter centrioles, decorates the external part of the microtubules along the CC with a 9-fold symmetry revealed from transversal sections (**Fig 1F**). The bulk of LCA5 lined MTDs in the bulge region and, to a lesser extent, the distal cilium, and was occasionally weakly present at centrioles and along the CC, where it might interact with FAM161A [10]. While LCA5 has been mostly described along the CC, its localization at the level of the bulge region corroborates previous immunogold results [22] (**Fig 1G**). We further analyzed the distribution of CEP290 relative to LCA5 and found that neither protein overlaps, with LCA5 lining up the MTDs in the extension of CEP290 on an approximately 1 μm long region (**S2 Fig**). We thus propose that LCA5 delineates the bulge region (**S1 Fig**), but whether other proteins, such as the microtubule-associated protein RP1 that has been described to be located more distally to the CC also decorate this region remains to be determined [23].

Next, to precisely map these different proteins relative to the MTDs within the ultrastructure of the CC, we measured the distances between peak intensities of POC5, CENTRIN, or CEP290 and tubulin from transversal section images, omitting FAM161A as the quality of the signal did not allow precise measurements. We found that POC5 and CENTRIN are internal to the microtubule wall by approximately 20 nm, similarly to their position within the inner scaffold at centrioles [16], while CEP290 is approximately 41 nm external to the MTDs (**Fig 1H**). By costaining CEP290 with the external tubulin glutamylation marker GT335, we also observed that CEP290 is contiguous to the MTDs (**Fig 1I and 1J**). To correlate these measurements with the CC ultrastructure, we then performed transversal EM sections of wild-type (WT) P14 mouse photoreceptors (**Fig 1K** and **S3 Fig**). By applying a 9-fold symmetrization to increase the contrast, we revealed the presence of the Y-link structures connecting the MTDs to the ciliary membrane, as well as an inner ring connecting neighboring MTDs, previously described as a transition zone ring [24,25] and reminiscent of the centriolar inner scaffold, which we therefore dubbed CC inner scaffold (**Fig 1K** and **S3 Fig**). By superimposing the measured distances of POC5, CENTRIN, and CEP290 onto a symmetrized EM image, we demonstrated that POC5 and CENTRIN perfectly colocalize with the CC inner scaffold and that CEP290 localizes at the level of the Y-links region (**Fig 1L**). To ascertain these results, we performed an U-ExM simulation from EM symmetrized images (**S4 Fig**) and confirmed that these structural elements depict similar localization signals to the mapped proteins. Taken together, these molecular and ultrastructural analyses confirmed the existence of an inner scaffold at the CC, linking MTDs together.

## Photoreceptor CC inner scaffold assembly

We then investigated how and when the CC, and notably the CC inner scaffold, assembles during postnatal development of photoreceptors in mouse, between days 4 and 60 (P4 and P60). We first monitored rod OS maturation at low magnification using tubulin and RHODOPSIN, the sensory protein that captures light and promotes its conversion into electric signals [26]. We found that U-ExM recapitulated OS formation with the appearance of discrete RHODOPSIN signals between P4 and P7 that gradually extend to form mature OS at P30 to P60 [3] (**Fig 2A**). Then, we analyzed at higher magnification the CC inner scaffold formation using POC5 signal as a proxy (**Fig 2B**). We found that at P4, while all centrioles are POC5-positive and axonemal microtubules are already elongating, only 42% of photoreceptors displayed POC5 staining at the level of the ciliary axoneme that may represent the nascent CC (**Fig 2B and 2K** and **S1 Fig**). Early afterward, at P7, POC5 is detected in 95% of the growing CC, spanning on average 415 nm (**Fig 2B, 2E and 2K**). The POC5 signal then elongates in parallel to OS maturation, until it reaches its final length of around 1,500 nm at P60, similar to the described length of a

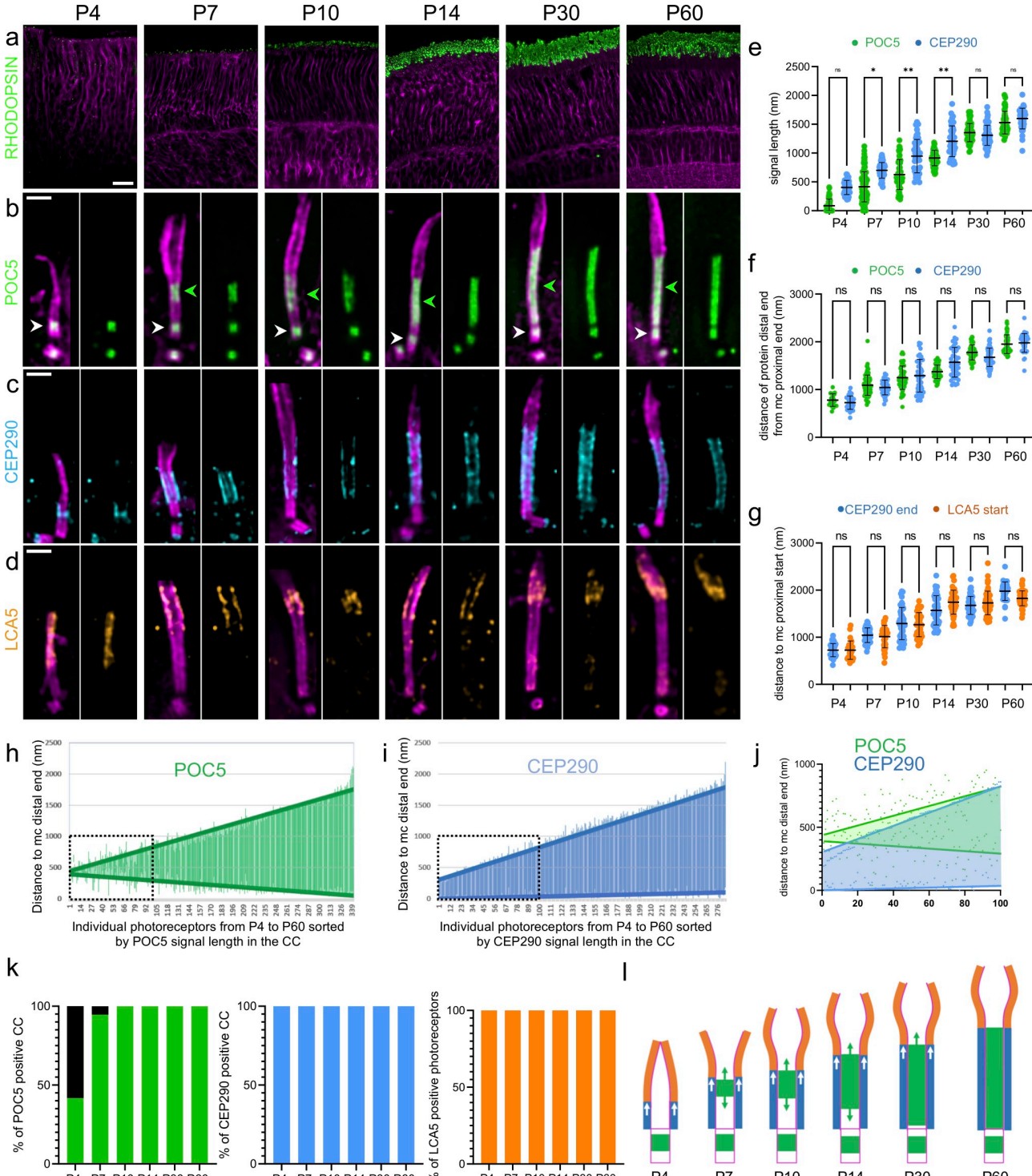

**Fig 2. Photoreceptor CC inner scaffold assembly.** (**a**) Low magnification of expanded retinas showing rod OS formation from P4 to P60 stained with RHODOPSIN (green) and tubulin (magenta). Scale bar: 20 μm. (**b–d**) Expanded photoreceptors stained for tubulin (magenta) and POC5 (green, b), CEP290 (cyan, c), or LCA5 (orange, d) from P4 to P60. Note that CEP290 also caps the daughter centriole. Green arrowheads = CC inner scaffold; white arrowheads = centriole inner scaffold. Scale bar: 500 nm. (**e**) Quantification of POC5 (green) and CEP290 (cyan) signal length inside the CC over time; ≥3 animals per time point. (**f**) Quantification of the distance of POC5 (green) and CEP290 (cyan) signal distal ends in the CC to the mother centriole proximal end from P4 to P60; ≥3 animals per time point. (**g**) Distance of CEP290 signal distal end (cyan) and LCA5 signal start (orange) to the mother centriole proximal end; ≥3 animals per time point. Note that CEP290 measurements are the same as in (f). (**h, i**) Length and position of POC5 (h: green) or CEP290 (i: cyan) signal within the CC. Note that signals are sorted by length and distances are compared to MC distal end (measurements from P4 to

P60). Thick green (POC5) and cyan (CEP290) lines represent linear regression curves. Black dashed square represents the inset highlighted in (j); ≥3 animals per time point. (**j**) Comparison of the 100 shortest CC reveals bidirectional POC5 (green) and unidirectional CEP290 (cyan) growth. (**k**) Percentage of CC positive for POC5 (green), CEP290 (cyan), and LCA5 (orange) from P4 to P60; ≥3 animals per time point. (**l**) Model showing inner scaffold (green) and CEP290 (cyan) growth over time, dictating the final length of the CC and the starting point of the bulge region (orange). Means, percentage, and standard deviations are listed in S1 Table. The data underlying all the graphs shown in the figure are included in the S1 Data file. CC, connecting cilium; OS, outer segment.

mature CC [27] (**Fig 2B** and **2E**). By measuring the relative position of the proximal and distal POC5 signal ends in the CC compared to the centriolar distal end, we also noticed that POC5 initially appears at variable distances from the mother centriole, while it appears below the beginning of the bulge region. We further found that its coverage elongates toward both the mature centriole and the distal end of the CC, suggesting that the CC inner scaffold structure growth is bidirectional (**Fig 2B, 2H, 2J and 2L** and **S5 Fig**).

We further investigated whether the CC inner scaffold assembly coincides with that of the Y-links and the bulge region defined by CEP290 and LCA5, respectively (**Fig 2C–2G**). We found that both CEP290 and LCA5 appear earlier than POC5 from P4 onwards, as 100% of photoreceptors displayed a signal for these 2 proteins (**Fig 2K**). In contrast to the bidirectional growth of POC5, CEP290 extends directly from the end of the underlying mother centriole and exhibits a unidirectional growth, indicating that the assembly of these 2 structural modules is probably independent (**Fig 2C, 2E, 2I, 2J and 2L** and **S5 Fig**). However, we noticed that the distal ends of the POC5 and CEP290 signals extend concomitantly and correspond to the starting point of LCA5 signal (**Fig 2F** and **2G**). This observation suggests that the growth of the CC region is dictated by the bidirectional elongation of the CC inner scaffold together with the unidirectional Y-links growth, and therefore displaces distally the beginning of the bulge region where membrane discs are formed (**Fig 2C–2G and 2L**).

## The CC inner scaffold acts as a structural zipper maintaining MTD cohesion

A key prediction of the CC inner scaffold function is that it bridges and maintains neighboring MTDs cohesion during its elongation. To test this hypothesis, we next explored the MTD diameter during CC formation. Indeed, while monitoring the growth of the CC inner scaffold, we observed that the axoneme diameter seemed wider in the absence of POC5, compatible with the fact that the absence of the CC inner scaffold coincides with spread MTDs (**Fig 3A**). To confirm this hypothesis, we measured between P4 and P60 the axoneme diameter at different positions: at the end of the POC5 signal (0), but also 150 nm below (−150) and above (+150) (**Fig 3B**). At every time point, we found a significant axoneme diameter enlargement distally to the end of the POC5 signal, suggesting that the CC inner scaffold gathers MTDs together early and throughout OS development (**Fig 3B**). Correspondingly, we found that the position of the microtubule enlargement corresponds to the distal end position of the CC inner scaffold marked by POC5 (**Fig 3C and 3D**). To test whether this microtubule spread is correlated with the absence of the CC inner scaffold at the ultrastructural level, we next analyzed EM transversal sections of P14 photoreceptors (**Fig 3E**). As expected, we found that the bulge region, identifiable by the absence of Y-links and displaying randomly shaped associated membranes, lacks the CC inner scaffold (**Fig 3E**). Moreover, this region has a larger perimeter and a less circular shape, correlating the lack of cohesion between MTDs and the absence of the Y-links (**Fig 3F and 3G**). The absence of microtubule cohesion was even more pronounced in the distal cilium region where singlet microtubules can be observed (**S1 Fig**). These results confirmed that the growing CC inner scaffold gathers neighboring MTDs during elongation, acting as a structural zipper during CC assembly.

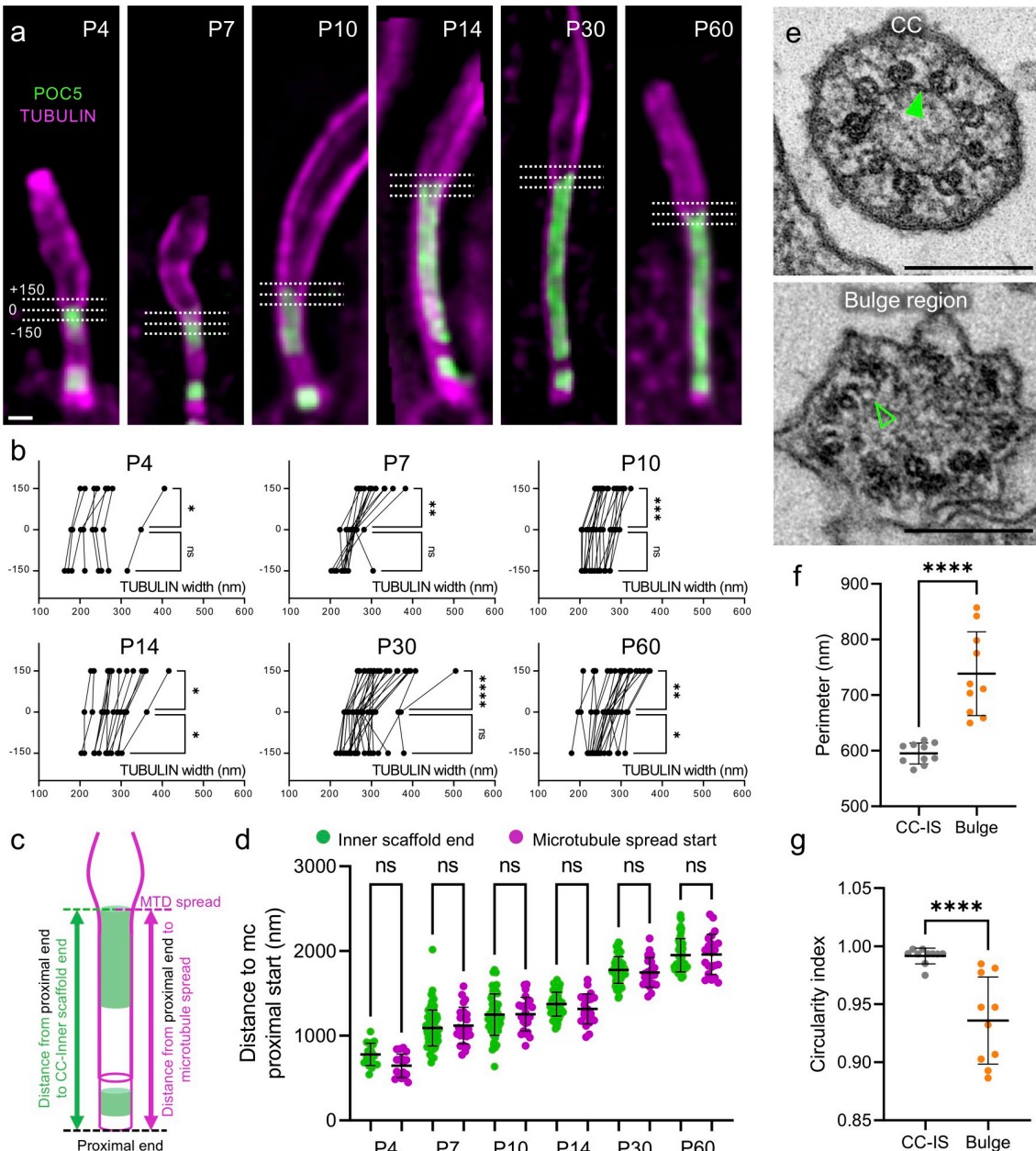

**Fig 3. The CC inner scaffold acts as a structural zipper maintaining MTDs cohesion.** (**a**) Expanded photoreceptors illustrating the measurements of tubulin width at 3 locations relative to the POC5 signal. 0: distal end of the CC inner scaffold; +150: 150 nm distally to the CC inner scaffold end; −150: 150 nm proximally to the CC inner scaffold end. Scale bar: 200 nm. (**b**) Tubulin width measurements of the photoreceptor at the 3 locations depicted in (a) (−150 nm, 0 nm, +150 nm) from P4 to P60; ≥3 animals per time point. (**c**) Scheme describing the measurements of the distances between mother centriole proximal end and CC inner scaffold signal end (green) or MTDs spread (magenta) used in (d). (**d**) Comparison of the position of the CC inner scaffold end (POC5) or microtubule spread start relative to the centriole's proximal end, from P4 to P60; ≥3 animals per time point. Note that inner scaffold measurements correspond to the POC5 data presented in Fig 2F. (**e**) EM transversal sections at the CC (top) or at the bulge region (bottom). Filled green arrowhead points to the CC inner scaffold; empty green arrowhead reveals the absence of the CC inner scaffold. Scale bar: 200 nm. (**f, g**) Distribution of the perimeter (f) or circularity (g) of the MTDs from transversal sections of photoreceptor CC (gray) or bulge (orange). *N* = 1 animal. Means and standard deviations are listed in S1 Table. The data underlying all the graphs shown in the figure are included in the S1 Data file. CC, connecting cilium; EM, electron microscopy; MTD, microtubule doublet.

## *Fam161a* mutation leads to the specific loss of the CC inner scaffold

Next, we tested whether the MTD spread observed in some retinal degenerative diseases such as RP [9] is linked to the structural loss of the CC inner scaffold and whether it precedes photoreceptor degeneration. For this purpose, we took advantage of a RP28 mouse model (*Fam161a*$^{tm1b/tm1b}$) deficient for FAM161A, which displays progressive photoreceptor degeneration accompanied by loss of visual function from 1 month of age and associated with spread MTDs [12].

First, using tubulin and RHODOPSIN, we assessed rod OS development from P4 to P60 in *Fam161a*$^{tm1b/tm1b}$ retinas and found no noticeable defects between P4 and P14, suggesting that early development of OS is not impacted (**Fig 4A** and **S6 Fig**). In contrast, from P30 onwards, RHODOPSIN signal progressively decreased, correlating with membrane disc disorganization and progressive loss of visual acuity in these animals [12]. Furthermore, tubulin signal in P60 reveals important defects with spread MTDs, recapitulating the phenotype previously seen in EM [9] (**Fig 4A–4D**). Similarly, we found that M/L OPSIN, a marker of the cone photoreceptor cells, was also strongly disorganized in *Fam161a*$^{tm1b/tm1b}$ retinas, with spread MTDs, indicating that cones are also degenerating in this model system (**S6 Fig**). As *Fam161a*$^{tm1b/tm1b}$ rods seem to develop normally until P14, we monitored potential residual FAM161A expression by immunostaining (**S7 Fig**). We found that a faint signal of FAM161A was still observable at the level of the CC in early time points with approximately 50% of CC positive until P14, suggesting that a truncated protein is still weakly expressed at early time points [12] (**S7 Fig**). In addition, we noticed that FAM161A seemed more stable at centrioles compared to the CC in *Fam161a*$^{tm1b/tm1b}$ mice, before vanishing with solely 12% of FAM161A positive centrioles at P30 (**S7 Fig**). We further validated this pattern of residual FAM161A signal in a second mouse model deficient for FAM161A (*Fam161a*$^{GT/GT}$), known to generate a truncated FAM161A protein [9] (**S7 Fig**). In this mouse model, the disappearance of FAM161A happened even earlier, with no more staining at either CC or centrioles at P14, suggesting that this mutant leads to a more severe phenotype than *Fam161a*$^{tm1b/tm1b}$ (**S7 Fig**).

We next monitored the localization of POC5 and CENTRIN during the development of photoreceptors in *Fam161a*$^{tm1b/tm1b}$ or *Fam161a*$^{GT/GT}$ retinas (**Fig 4B** and **S7 Fig**). Similarly to FAM161A, POC5 and CENTRIN proteins showed an early and transitory localization at the CC before disappearing later on in the majority of the photoreceptors observed (**Fig 4B, 4F and 4I** and **S7 Fig**). However, mutant animals were heterogeneously impacted over time, exemplified by the distribution of POC5 signal length within the CC between P14 and P60, explaining differences observed between POC5 and FAM161A stainings for late time points (**Fig 4I** and **S7 Fig**). This loss is paralleled by an opening of the axonemal MTDs that progressively reaches the mother centriole distal end, indicative of a shortening of the CC region, as previously described [9] (**Fig 4B, 4E** and **4F**). This MTD spread becomes prominent at P30, where OS start to be impacted, accounting for the beginning of visual function decline at this age [12]. In contrast to FAM161A, we noticed that POC5 and CENTRIN remain present at centrioles throughout the analyzed time points, suggesting a differential impact of FAM161A mutation on CC and centriole inner scaffold structures that could explain the exclusive retinal phenotype observed in RP28 (**Fig 4B** and **S7 Fig**). To further understand this result, we monitored POC5 in cycling human U2OS cells depleted of FAM161A using small interfering RNA (siRNA). While FAM161A was absent in 88% of centrioles in FAM161A-depleted cells, only 38% of centrioles had lost POC5 (**S8 Fig**), suggesting that FAM161A does not control POC5 recruitment at centrioles, consistent with the results obtained in photoreceptors. In contrast, we found that FAM161A was reduced similarly to POC5 at centrioles in POC5-depleted U2OS cells (**S8 Fig**), revealing that POC5 mostly controls FAM161A localization at centrioles.

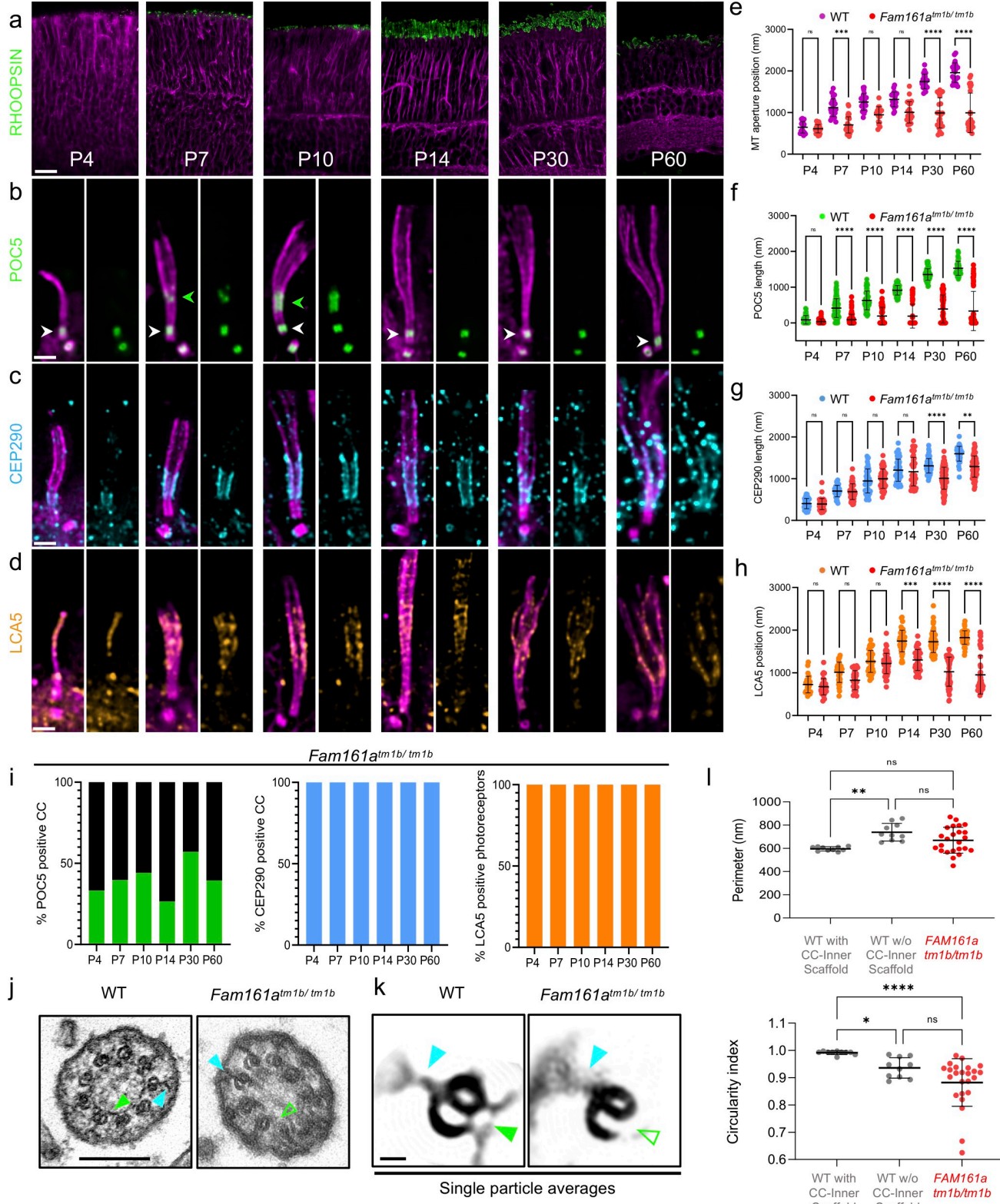

**Fig 4. *Fam161a* mutation leads to the specific loss of the CC inner scaffold.** (**a**) Low magnification of expanded *Fam161a^{tm1b/tm1b}* retinas stained for RHODOPSIN (green) and tubulin (magenta) from P4 to P60. Scale bar: 20 μm. (**b–d**) Expanded *Fam161a^{tm1b/tm1b}* photoreceptors stained for POC5 (green, b), CEP290 (cyan, c), and LCA5 (orange, d) from P4 to P60. Note the transitory appearance of the CC inner scaffold between P7 and P10 (green arrowheads) followed by its collapse, paralleling microtubule spread. In contrast, centriole inner scaffold (white arrowheads) is retained over time. Scale

bar: 500 nm. (**e**) Comparison of the microtubule enlargement position relative to MC proximal end between WT (magenta) and *Fam161a^{tm1b/tm1b}* (red) photoreceptors; ≥3 animals per time point. (**f–h**) Impact of the *Fam161a^{tm1b/tm1b}* mutant on CC inner scaffold length (POC5 staining, f), CEP290 length (g), or LCA5 start position (h); ≥3 animals per time point. (**i**) Percentage of CC positive for POC5 (green), CEP290 (cyan), and photoreceptor positive for LCA5 (orange) from P4 to P60 in *Fam161a^{tm1b/tm1b}* photoreceptors; ≥3 animals per time point. Note that the heterogeneity is found between animals and not within a single animal. (**j**) EM micrographs of WT (left) or *Fam161a^{tm1b/tm1b}* (right) CC transversal sections. Scale bar: 200 nm. (**k**) single particle averages of WT or *Fam161a^{tm1b/tm1b}* MTDs. Note that Y-links are present in both conditions (cyan arrowheads), whereas CC inner scaffold is present in WT (full green arrowhead) but absent in *Fam161a^{tm1b/tm1b}* (empty green arrowhead). Scale bar: 20 nm. (**l**) Comparison of the perimeter or the circularity of the microtubule axoneme from transversal sections of WT (gray) or *Fam161a^{tm1b/tm1b}* (red) photoreceptors. Note that in panels e–h and l, WT measurements are the same as presented in Fig 2E–2G and Fig 3D, 3F and 3G. Means, percentage, and standard deviations are listed in S1 Table. The data underlying all the graphs shown in the figure are included in the S1 Data file. CC, connecting cilium; EM, electron microscopy; MTD, microtubule doublet; WT, wild type.

Finally, codepletion of both proteins strongly affects FAM161A and POC5 presence at centrioles and induces centriole architectural defects with broken microtubule pieces in approximately 15% of cases, a phenotype reminiscent of the spread MTDs in the connecting cilium in *Fam161a* mutants (**S8 Fig**).

We also investigated whether CEP290 and LCA5 are affected in *Fam161a^{tm1b/tm1b}* photoreceptors (**Fig 4C** and **4D**). In contrast to POC5, 100% of photoreceptors displayed CEP290 or LCA5 signals over the analyzed time points, reinforcing the independence of these different modules (**Fig 4I**). We observed a decrease in CEP290 length from P30 onwards, presumably as a secondary consequence of MTD defects affecting axoneme integrity at this time point (**Fig 4C** and **4G**). LCA5 localization was also not impaired at early time points, but it re-localized closer to the mother centriole from P14, consistent with the shortening of the CC and revealing the disorganization of the bulge region, possibly causing membrane disc formation impairment (**Fig 4D** and **4H**). We next analyzed SPATA7, a putative CEP290 interactor described as maintaining the distal portion of the CC [28]. First, analyzing WT mature photoreceptors, we confirmed that SPATA7 decorates the entire CC length (**S9 Fig**). Despite a low signal quality that precluded precise transversal localization, SPATA7 signal lay mostly outside the MTDs. Interestingly, in P60 *Fam161a^{tm1b/tm1b}* photoreceptors, SPATA7 signal greatly resembles CEP290, being still present and following each spread MTD, suggesting that this protein could exhibit a 9-fold symmetry and be part of the CEP290 module (**S9 Fig**).

Altogether, these results show that a specific loss of CC inner scaffold proteins, which initially localize transitorily in *Fam161a^{tm1b/tm1b}* retinas and presumably allow early OS formation, triggers MTDs destabilization and CC shortening, prior to OS degeneration. We next investigated whether this loss of CC inner scaffold proteins is correlated with a loss of the CC inner scaffold structure, by analyzing transversal EM sections of a *Fam161a^{tm1b/tm1b}* P30 mouse retina (**Fig 4J** and **S10 Fig**). We found that the vast majority of the axonemal sections lacked CC inner scaffold, whereas Y-links were still present, confirming the molecular architecture findings made by U-ExM. Moreover, the global MTD structure was sometimes greatly impaired, with opened B-microtubules and with membrane invaginations found inside the axoneme (**S10 Fig**). In keeping with this, axonemal perimeter distribution was largely heterogeneous in mutant photoreceptors, together with a less round shape, demonstrating that the lack of the CC inner scaffold impairs MTD cohesion (**Fig 4L**). To verify this lack of CC inner scaffold, we performed particle averaging on WT and *Fam161a^{tm1b/tm1b}* MTDs from EM micrographs and highlighted the underlying structural defects (**Fig 4K** and **S11 Fig**). This analysis confirmed the presence of the Y-links as well as the complete structural loss of the CC inner scaffold in *Fam161a^{tm1b/tm1b}*.

## Loss of the CC inner scaffold causes photoreceptor degeneration

Finally, we studied the effect of the CC inner scaffold loss on overall photoreceptor cell integrity and, in particular, the rod outer segment membrane stacks containing RHODOPSIN

(**Fig 5A–5D**). We found that while RHODOPSIN is correctly located above the CC in P60 WT cells, the loss of CC inner scaffold induces a collapse of the OS structure, with the RHODOPSIN signal and membrane discs displaced toward and inside the CC region (**Fig 5B and 5D** and **S10 Fig**). Moreover, in addition to a mislocalization of RHODOPSIN at the cell body, as previously observed [9] (**S12 Fig**, white arrows), we noticed an accumulation at the base of the

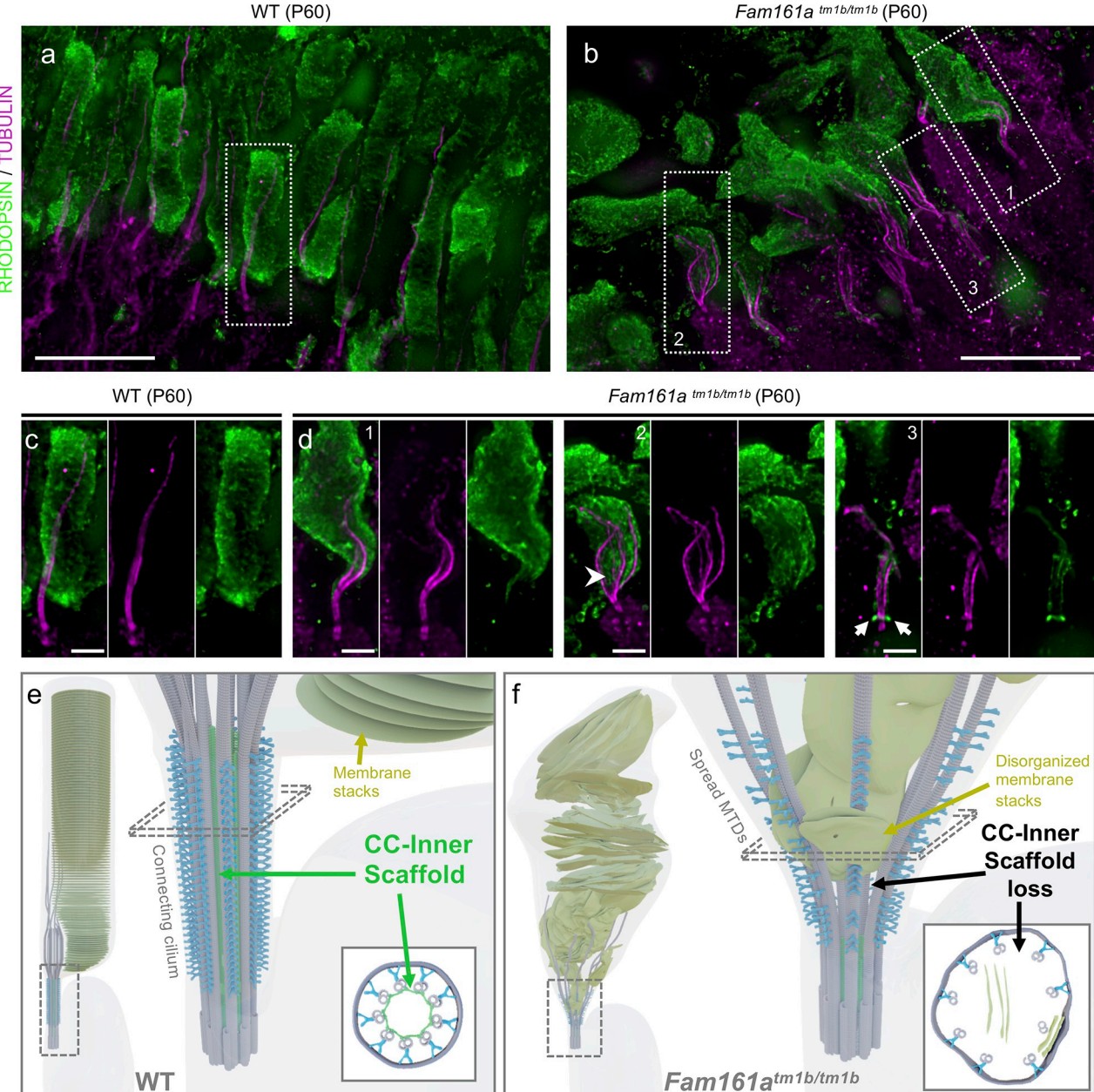

**Fig 5. Loss of the CC inner scaffold causes photoreceptor degeneration.** (**a, b**) Low magnification of expanded WT (a) or *Fam161a*$^{tm1b/tm1b}$ (b) retinas stained for RHODOPSIN (green) and tubulin (magenta) at P60. Note the microtubule defects in the mutant, accompanied by outer segment collapse. Dashed lines represent the insets (1, 2, and 3) depicted in (c) and (d). Scale bar: 5 μm. (**c, d**) Insets from WT (c) or *Fam161a*$^{tm1b/tm1b}$ (d) retinas depicted in (a) and (b), respectively. Arrowhead shows the RHODOPSIN signal entering inside the microtubules. Arrows depict the accumulation of the RHODOPSIN at the base of the CC. Scale bar: 1 μm. (**e, f**) Model representing mature WT (e) or *Fam161a*$^{tm1b/tm1b}$ (f) photoreceptor with the CC region highlighted. Schemes representing transversal sections are depicted in the bottom right of each panel. CC, connecting cilium; WT, wild type.

CC in several cells (**Fig 5D,** white arrows), potentially indicating a defect in RHODOPSIN transport in *Fam161a^{tm1b/tm1b}*. These results demonstrate that CC inner scaffold acts as a structural foundation to maintain the cell architecture and a correct outer segment organization.

## Discussion

The architecture of photoreceptor cells is unique, notably because of the presence of the large outer segment cilium that may represent the most highly specialized primary cilium that can be found in the human body. This towering cellular structure, extending up to 50 μm in length [29] and 2.5 μm in diameter [30], is connected to the rest of the cell body by the CC, a thin region of 1.5 μm long and 200 nm in width. Despite its relatively short size, the CC represents the longest transition zone-like structure observed so far, with Y-links and ciliary necklace distributed all along its length, allowing gating control for ciliary entry and trafficking [25]. Given these characteristics and its central position in photoreceptor cell function, it is not surprising that the CC has been identified over the years as a hotspot for proteins whose mutations are associated with photoreceptor degeneration [31]. However, despite the growing number of retinopathy-associated proteins that have been described at the level of the CC, their precise roles inside this compartment are mostly not known. One of the reasons is the limited resolution of fluorescence imaging methods that limit our ability to correlate protein location with the CC ultrastructure. Here, using for the first time super-resolution U-ExM expansion microscopy on retina tissue, we were able to reveal an unprecedented nanoscale molecular mapping of the mammalian connecting cilium, an approach that opens new avenues to better understand the biology of photoreceptor cells. In addition, correlating these results with CC ultrastructure observed in electron microscopy (EM) allowed us to highlight the presence and the composition of a previously poorly described structure: the connecting cilium inner scaffold. This structural element, lining the inner wall of the microtubule doublets all along the connecting cilium and composed of the core proteins POC5, CENTRIN, and FAM161A, appears to be highly similar to the recently described centriole inner scaffold [16]. Interestingly, the centriole inner scaffold has been proposed to ensure centriole integrity by maintaining a tight cohesion between microtubule triplets [17]. Here, we demonstrate that the CC inner scaffold might share a similar function by bridging neighboring microtubule doublets, acting as a structural zipper, thus maintaining microtubule doublet cohesion in the CC region. Consistent with this model, we showed that depletion of the inner scaffold protein FAM161A causes the structural loss of the CC inner scaffold accompanied by the loss of cohesion of MTDs inside the CC. This axonemal disorganization leads to the collapse of the whole outer segment, presumably causing photoreceptor degeneration (**Fig 5**). Importantly, we associated the absence of a CC protein with the structural loss of the CC inner scaffold, allowing us to propose a molecular and structural mechanism explaining this subtype of retinitis pigmentosa. Furthermore, we also noticed by EM that the loss of the CC inner scaffold is often accompanied by an opening at the level of the B-microtubule, suggesting that depletion of FAM161A could lead to the loss of the stem, an inner scaffold structural subcomponent that connects the microtubule doublet at the level of the A to B inner junction [16].

It remains unclear how loss of the CC inner scaffold induces late photoreceptor degeneration. One could imagine that a large-sized outer segment, with hundreds of membrane stacks and an estimated 40 million rhodopsin proteins [32], needs a robust foundation to avoid its collapse. Therefore, as the CC inner scaffold starts to form transiently during outer segment formation, no defects are observed at early time points of photoreceptor development in mutant mice. In contrast, the lack of the CC inner scaffold later on induces microtubule

doublet spreads and disorganization of the outer segment that will ultimately lead to photoreceptor death.

Another interesting question raised by the discovery of CC inner scaffold is its possible conservation in other axonemal structures or species, which have not yet been characterized as such. For instance, it has been shown that an ortholog of *FAM161A*, FAM-161, is conserved in the nematode *Caenorhabditis elegans*, and that this protein decorates a long transition zone structure at the ciliary base in sensory neurons [33]. It has been shown by EM that this region is composed by an inner ring structure connecting microtubule doublets [34], which appears very similar to the CC inner scaffold. These 2 independent observations suggest that an inner scaffold structure may be conserved in *C. elegans* sensory neurons. Further research to examine its evolutionary conservation in different models would not only be important to better understand the specific role of internal scaffolding structures in different cilia, but also their evolutionary origin.

Our study has also shown that, despite its evident similarities with the centriole inner scaffold, CC inner scaffold formation during photoreceptor outer segment maturation does not emanate or elongate from the underlying centriole inner scaffold, suggesting that the formation of these 2 similar structures is independent. Even once fully formed, a gap between the centriole IS and the CC inner scaffold remains (Fig 1), suggesting that they do not form a continuous structure. This was also confirmed using a mouse model of *Fam161a* mutation, where the CC inner scaffold was affected and the centriole inner scaffold unimpaired, in agreement with the phenotype observed, where only the eyes are affected. Further work using cryo-electron tomography will be helpful to fully understand the architecture of the CC inner scaffold and potentially highlight subtle differences between the centriole and CC inner scaffold structures, explaining why centrioles appear unaffected by this mutation.

The CC has always been considered as a long transition zone (TZ) specific to the photoreceptor cells, notably because of the observation in the 70's of 2 TZ structures (Y-links and ciliary necklace) all along the CC [25]. Since then, different TZ proteins have been described as decorating the CC in photoreceptor cells, corroborating this observation. For example, the protein CEP290 has been proposed to be part of the Y-links structure, even if this is still a matter of debate in the literature [20,27,35]. Here, we showed for the first time the 9-fold symmetry of the CEP290 signal all along the CC, and positioned it at the level of the Y-links structure. Moreover, we demonstrated that CEP290 setup and maintenance in the CC are independent of the CC inner scaffold, revealing that different modules can be discriminated inside the CC. Consistently, Potter and colleagues showed that *Cep290*-knockout photoreceptors still possess the CC inner scaffold protein CENTRIN, suggesting that CEP290 and the CC inner scaffold are probably independent [20]. These results suggest that the CC might differ from a long canonical TZ, as previously suggested. In line with this, it has recently been proposed that the CC could actually be partitioned in 2 subregions that could recruit different proteins [28]. Therefore, fine investigation of CC composition with U-ExM, but also of how transport is mediated through this long connecting region, would be of great interest for the field.

Finally, U-ExM allowed us to gain new insights into the "bulge" region, where the crucial process of membrane stack formation occurs. We revealed the enrichment of LCA5 protein within this compartment, lining microtubules with a 9-fold symmetry. While we found LCA5 mostly enriched at the bulge region, we also found some weak localization at the level of the connecting cilium and the centriole that might explain the reported interaction between LCA5 and FAM161A. However, further investigations might be needed to elucidate this interaction at the level of the CC. Given LCA5's suggested association with the transport machinery in photoreceptor cells [36], U-ExM could now help to decipher molecular processes in this region that direct new membrane formation and microtubule elongation.

In summary, our work highlights the presence of an inner scaffold structure inside the connecting cilium, containing FAM161A, POC5, and CENTRIN, that is crucial for proper maintenance of the photoreceptor OS in mouse (**Fig 5E and 5F**). Moreover, this work suggests a molecular and structural basis to understand photoreceptor degeneration in retinitis pigmentosa linked to defective cilia, paving the way for future therapeutic options to restore visual acuity in RP patients.

## Ethic statement

All animal experiments were conducted with the authorization numbers VD1367, according to the guidelines and regulations issued by the Swiss Federal Veterinary Office.

## Material and methods

### Mouse model

Animals were handled in accordance with the statement of the "Animals in Research Committee" of the Association for Research in Vision and Ophthalmology, and experiments were approved by the local institutional committee (VD1367). The mice were maintained at 22˚C with a 12-h light/12-h dark cycle with light on at 7:00 AM and were feed ad libitum. *Fam161a*-deficient mice, *Fam161a*$^{tm1b/tm1b}$ [12] and *Fam161a*$^{GT/GT}$ [9], were obtained from Avigail Beryozkin and Thomas Langmann, respectively. C57BL6RJ (Janvier Labs, Le Genest-Saint-Isle, France) were used as controls.

Mice were sacrificed at postnatal day P4, P7, P10, P14, P30 (1 month), and P60 (2 months) of age and eyes enucleated.

### Retina dissection

After enucleation, eyes were fixed for 15 min at room temperature (RT) in 4% PFA (paraformaldehyde, P6148, Sigma-Aldrich) in phosphate-buffered saline (PBS) (P4, P7, P10, P14) or 2% PFA in PBS (1 month and 2 months) and then transferred into (PBS). Then, cornea and lens were removed with micro scissors, and the sclera was separated from the retina. Retinas were then either kept as a cup for EM processing or incised to flatten it as a clover inside a 10-mm microwell of a 35-mm petri dish (P35G-1.5-10-C, MatTek) to allow their processing by Ultrastructure expansion microscopy (U-ExM).

### Retina expansion

Protocol for retina expansion was adapted from the U-ExM method [18] with few optimizations. Briefly, crosslinking prevention step was extended to overnight (ON) incubation with 100 μL of 2% acrylamide (AA; A4058, Sigma-Aldrich) + 1.4% formaldehyde (FA; F8775, Sigma-Aldrich) at 37˚C inside the 10-mm microwell of a petri dish (MatTek). Then, the solution was removed and 35 μL monomer solution (MS) composed of 25 μl of sodium acrylate (stock solution at 38% (w/w) diluted with nuclease-free water, 408220, Sigma-Aldrich), 12.5 μl of AA, 2.5 μl of N, N′-methylenbisacrylamide (BIS, 2%, M1533, Sigma-Aldrich), and 5 μl of 10× PBS was added for 90 min at RT to allow its penetration into the tissue. Then, MS was removed and 90 μL of MS was added together with ammonium persulfate (APS, 17874, Thermo Fisher Scientific) and tetramethylethylenediamine (TEMED, 17919, Thermo Fisher Scientific) as a final concentration of 0.5% for 45 min at 4˚C first followed by 3 h incubation at 37˚C to allow gelation. A 24-mm coverslip was added on top to close the chamber. Next, the coverslip was removed and 1 ml of denaturation buffer (200 mM SDS, 200 mM NaCl, 50 mM Tris Base in water (pH 9)) was added into the MatTek dish for 15 min at RT with shaking.

Then, careful detachment of the gel from the dish with a spatula was performed, and the gel was incubated in 1.5 ml tube filled with denaturation buffer for 1 h at 95°C and then ON at RT. The day after, the gel was cut around the retina that is still visible at this step, and expanded in 3 successive ddH2O baths. Then, the gel was manually sliced with a razorblade to obtain approximately 0.5 mm thick transversal sections of the retina that were then processed for immunostaining.

## Human cell culture and expansion

Human U2OS were cultured similarly to [17]. Briefly, cells were grown in DMEM supplemented with GlutaMAX (Thermo Fisher Scientific), 10% fetal calf serum (Thermo Fisher Scientific), and penicillin and streptomycin (100 μg/ml, Thermo Fisher Scientific). Expansion microscopy of U2OS cells was performed as previously described [16].

## Immunostaining

After expansion, human cell gels were shrunk with three 5-min baths of 1× PBS. Then, primary antibodies were incubated for 3 h at 37°C in PBS with 2% of bovine serum albumin (BSA). Gels were washed 3 times 10 min in PBS with 0.1% Tween 20 (PBST) prior to secondary antibodies incubation for 3 h at 37°C. After a second round of washing (3 times 10 min in PBST), gels were expanded with three 30-min baths of ddH20 before imaging. Antibodies used are referenced in S2 Table. Tubulin staining presented in the figures corresponds to either a mixture of anti α- and β-tubulin (raised in mouse) or anti α-tubulin alone (raised in rabbit), depending on the species of the other protein stained.

For retina slices immunostaining, primary antibody incubation was prolonged overnight at 4°C. Image acquisition was performed on an inverted confocal Leica TCS SP8 microscope or on a Leica Thunder DMi8 microscope using a 20× (0.40 NA) or 63× (1.4 NA) oil objective with Lightning or Thunder SVCC (small volume computational clearing) mode at max resolution, adaptive as "Strategy" and water as "Mounting medium" to generate deconvolved images. 3D stacks were acquired with 0.12 μm z-intervals and an x, y pixel size of 35 nm.

## CEP290 protein and monoclonal antibody production

A fragment of the human *CEP290* cDNA encoding amino acids (1 to 299) was cloned into pGEX4T2 and expressed in *Escherichia coli* BL21 pRIL as a GST fusion protein.

Recombinant CEP290 (1–299) was produced in *E. coli* using 0.5 mM IPTG to induce expression over 3 h at 37°C. Immobilized glutathione (GST-Bind Resin, Novagen) was used for the purification of the recombinant GST fusion protein from bacterial cells. Beads were washed 3 times in GST Binding Buffer (100 mM HEPES (pH 7.3), 150 mM NaCl) supplemented with Protease inhibitor tablets (c0mplete Mini, EDTA-free, Roche), PMSF, lysozyme, and DTT (Buffer A), followed by pelleting at 500 *g* at 4°C. The 20-l bacterial culture was harvested by centrifugation for 15 min at 6,000 *g* and the pellets frozen at −20°C. Pellets were resuspended in 20 ml of Buffer A. Tubes were rotated at 4°C for 30 min, then the cell lysate was sonicated 6 times for 5 s (total time of 30 s) with a 5 s pause between each pulse, at 20% amplitude, using a Digital Sonifier (Branson, London, United Kingdom) to solubilize proteins and fragment chromosomal DNA. Triton X-100 was then added to a final concentration of 1% and the sample incubated for 5 min at 4°C with gentle rocking to reduce nonspecific protein interactions. The extract was centrifuged at sixteen 100 *g* for 15 min at 4°C to remove cell debris. The supernatant was mixed with the previously equilibrated beads. A total of 1 ml bed volume of GST resin per liter of lysate was used. Lysate and beads were then incubated on a rotating mixer for 1 to 2 h at 4°C to capture GST fusion proteins. After binding, the beads

were washed 3 times (once with Buffer A, once with GST Binding Buffer supplemented with 0.5% Triton X-100, and finally with Tris–HCl 50 mM, $CaCl_2$ 10 mM (pH 8.0)) by spinning at 500 $g$ at 4°C. Thrombin linked to agarose beads (Sigma Clean Cleave kit) (1 ml of slurry used per 4 liters of bacterial culture) were washed 3 times with Tris–HCl 50 mM, $CaCl_2$ 10 mM (pH 8.0) by spinning at 500 $g$ at 4°C, then added to the washed GST resin.

The fusion protein was cleaved from the GST tag by incubation with thrombin for 16 to 18 h at 37°C. After incubation, GST beads and thrombin-agarose beads were pelleted at 16,000 $g$ for 5 min at 4°C and the supernatant containing the protein was transferred to a clean micro-centrifuge tube and kept on ice. Protein concentration was measured with a Bradford assay, and samples were stored at −80°C. Samples were taken at each step of this procedure for SDS-PAGE analysis, together with previously collected uninduced and induced samples. Gels were stained with Instant Blue Coomassie. Frozen protein samples were sent on dry ice for mouse immunization and hybridoma production (Cambridge Research Biochemicals). After testing by immunoblot and subcloning, clone 1C3G10, which produces IgG2a, was selected and expanded.

### Human cell transfection

U2OS cells were plated onto coverslips in a 6-well plate at 100,000 cells/well 24 h before transfection. Using Lipofectamine RNAimax (Thermo Fischer Scientific), cells were transfected either with 50 nM of silencer select negative control siRNA1 (4390843, Thermo Fisher Scientific), 25 nM of siRNA against POC5, or a mixture of 12.5 nM of 2 different siRNAs against FAM161A. For double transfections, a mixture of 25 nM POC5 siRNA with both FAM161A siRNAs at 12.5 nM each was used. Medium was changed 6 h post transfection, and cells were analyzed 72 h after transfection. siRNA references and sequences are referenced in S2 Table.

### Electron microscopy

For EM analyses, retina cups were first incubated overnight at RT with 3% PFA (15710, Electron Microscopy Sciences) and 0,1% glutaraldehyde (16200, Electron Microscopy Science) in PBS. Samples were further treated with 2% osmium tetroxide (05500-1$g$, Sigma-Aldrich) in buffer for 30 min and immersed in a solution of uranyl acetate (21447-25$g$, Polysciences) 0.25% overnight to enhance contrast of membranes. Samples were dehydrated in increasing concentrations of ethanol followed by pure propylene oxide (82320-1L, Sigma-Aldrich), and then embedded in Epon resin. Serial ultrathin sections of 50 or 100 nm were finally cut and stained with 5% uranyl acetate (in H2O) and Reynolds' lead citrate [37]. Micrographs of the WT sample were acquired using a G2 Sphera microscope operated at 120 kV equipped with an Eagle detector at a magnification of 25,000× corresponding to a pixel size 4.5Å. The micrographs of the mutant sample were acquired using a Talos LC120 microscope operated at 120 kV equipped with a CetaD detector at a magnification of 36,000× corresponding to pixel size 4 Å. All micrographs were acquired with a defocus between −3 to −5 μm. Location of the different sections along the proximal distal axis was determined thanks to the shape of the plasma membrane and the presence of different structures (Y-Links, Inner scaffold). CC has round-shaped membranes, inner scaffold, and Y-links, whereas the bulge area has no Y-links nor inner scaffold and has a random membrane shape (due to the absence of the Y-links and the nascent OS disk formation).

### Symmetrization

Symmetrization of EM micrographs was done using CentrioleJ plugin [38]. The first step consists of the circularization of the pattern (here the CC), to correct elliptical deformation of the

acquisition, by manually picking all the center of mass of the A-microtubules. Then, symmetrization consists in rotating the source image according to its symmetry (here 9-fold) and sum-projecting all the rotated images.

## Particle classification and averaging

A total of 745 and 178 particles were manually picked on the microtubule doublets and extracted with box size of 310 and 256 pixels from WT and mutant, respectively. The particles were subjected to 2D classification procedure implemented in Cryosparc 3.2 [39]. The alignment resolution and maximum resolution were limited to 50 and 10 Å, respectively, for alignment over 40 iterations. Moreover, Force Max over poses/shifts, Enforce nonnegativity and use clamp-solvent to solve 2D classes were set to true to reduce background noise.

## Quantifications

**Expansion factor.** The expansion factor of each experiment was calculated in a semiautomated way by comparing the full width at half maximum (FWHM) of photoreceptor mother centriole proximal tubulin signal with the proximal tubulin signal of expanded human U2OS cell centrioles using PickCentrioleDim plugin described elsewhere [40]. Briefly, for each experiment, at least 10 photoreceptor mother centrioles FWHM were measured and compared to a pre-assessed value of U2OS centriole width (25 centrioles: mean = 231.3 nm +/− 15.6 nm). The ratio between measured FWHM and known centriole width gave the expansion factor.

**Protein shift.** To calculate the protein shift compared to tubulin, confocal top view images of connecting cilia were analyzed. Using ImageJ, a line crossing connecting cilia on their diameter was drawn and plot profiles of each channel (protein of interest and tubulin) were generated. Then, distances between peak intensities of the protein of interest and tubulin were measured and corrected with the mean expansion factor calculated from all experiments (**S1 Fig**). For each connecting cilium, 4 measurements were taken to correct potential tilting effects.

**Protein signal length and position.** Protein signal lengths or position compared to mother centriole proximal end (depicted with tubulin) were measured using a segmented line drawn by hand (ImageJ) to fit with photoreceptor curvature, and corrected with the expansion factor. Only photoreceptors where both protein signal (POC5, CEP290, or LCA5) and centriole proximal end (tubulin) were clearly visible were selected for measurement.

**Tubulin spread.** Tubulin spread was assessed at each time point in a semiautomated way by measuring FWHM of tubulin signal with PickCentrioleDim plugin [40] on 3 different locations of the photoreceptor corresponding to 150 nm proximally to the end of the CC POC5 staining (−150), at the level of the end of the CC POC5 staining (0), or 150 nm distally to the end of the CC POC5 staining (+150). Each measurement was subsequently corrected with its respective expansion factor. The position of the microtubule opening was measured manually using a segmented line drawn in ImageJ [41], to fit with photoreceptor curvature, and corrected with the expansion factor. Independent tubulin staining alone was used to avoid bias with CC inner scaffold staining.

**Microtubule axoneme perimeter and circularity.** Microtubule axoneme perimeter and circularity were assessed from EM images. Using ImageJ, elliptical lines were drawn by hand to fit with microtubule doublets center allowing the measurements of the perimeter and the area. Circularity (Ci) was calculated using the formula: $Ci = 4^*Pi^*Area/Perimeter^2$. Only axonemes with all the microtubule doublets visible were quantified. Bulge regions were defined as lacking the inner scaffold ring and the Y-link structures.

**siRNA.** In order to evaluate siRNA efficiency after transfection, POC5 and FAM161A stainings after U-ExM were first assessed by eye by measuring the proportion of centrosomes (mother and daughter centrioles) with a complete signal at both centrioles (nondepleted) and centrosomes with 1 or both centrioles with a reduced or absent signal for the proteins (depleted).

For fluorescence intensity measurements of FAM161A and POC5, maximal projections were used using Fiji [41] on non-deconvoluted images. The same circular region of interest (ROI) drawn by hand was used to measure protein signal intensities around every centriole and their corresponding background. Fluorescence intensity was finally calculated by subtracting the average value of 2 background measures (raw integrated density).

**Statistical analyses.** The comparison of 2 groups was performed using nonparametric Mann–Whitney test, if normality was not granted because rejected by Pearson test. The comparisons of more than 2 groups were made using nonparametric Kruskal–Wallis test followed by post hoc test (Dunn's for multiple comparisons) to identify all the significant group differences. Every measurement was performed on at least 3 different animals, or cell culture independently, unless specified. Data are all represented as a scatter dot plot with centerline as mean, except for percentages quantifications, which are represented as histogram bars. The graphs with error bars indicate SD (+/−) and the significance level is denoted as usual (*$p < 0.05$, **$p < 0.01$, ***$p < 0.001$, ****$p < 0.0001$). All the statistical analyses were performed using Prism9. Every mean, percentage, standard deviation, test, and the number of animals used for comparison are referenced in S1 Table. When possible, a minimum of 10 measurements has been performed per animal. The data underlying the graphs shown in all the figures are included in the S1 Data file.

## Supporting information

**S1 Data. Raw and statistical data analysis.**
(XLSX)

**S1 Table. Descriptive statistics for all measurements.**
(XLSX)

**S2 Table. Antibodies and siRNA references.**
(XLSX)

**S1 Fig. U-ExM protocol adapted to mouse retina.** (a) Scheme summarizing the main steps of mouse retina dissection and expansion. (b) Expanded photoreceptor (left) highlighting the different regions of the distal inner segment and outer segment provided by the tubulin staining, together with the corresponding EM images (right). Scale bars: left = 500 nm; right = 200 nm. Black arrowhead points to microtubule singlet in the distal cilium. Note that the EM picture depicting the bulge region is the same as in the main figure. (c) Photoreceptor expansion factor calculation by comparing tubulin signal width at the photoreceptor mother centriole proximal end with the tubulin width of U2OS centriolar proximal end. Scale bars: 200 nm. (d) Fluorescence plot profile of tubulin at the level of the photoreceptor mother centriole proximal end generated with PickCentrioleDim plugin. (e) Measurements of U2OS centriole proximal width. $N = 4$ independent experiments. See S1 Table. (f) Distribution and mean of the expansion factors calculated from all the experiments performed on retinas. Note that developmental stage has no impact on the gel expansion; >40 experiments were used to measure expansion factor. See S1 Table. (g) Representative images of P4 photoreceptors with or without the CC inner scaffold. Scale bar: 200 nm. The data underlying all the graphs shown in the figure are included in the S1 Data file. CC, connecting cilium; EM, electron microscopy; FWHM, full

width at half maximum.
(TIFF)

**S2 Fig. LCA5 lines the MTDs in the extension of CEP290.** (a) Representative images of photoreceptor outer segments stained for CEP290 (green) and LCA5 (magenta). Dotted line rectangle represents an example of the region used in (c) for plot measurements. Scale bar: 500 nm. (b) Confocal image of a photoreceptor outer segment stained for CEP290 (green) and LCA5 (magenta). Scale bar: 500 nm. (c) Graph representing the average normalized intensity of CEP290 (green) or LCA5 (magenta) along the proximal to distal axis from 11 different pictures where examples are depicted in (a). Standard deviation is represented with transparent green (CEP290) or magenta (LCA5) areas. (d) Quantification of LCA5 signal length at the level of the bulge region in mature photoreceptors ($\geq$P60). $N = 3$ animals. The data underlying all the graphs shown in the figure are included in the S1 Data file. MTD, microtubule doublet; WT, wild type.
(TIFF)

**S3 Fig. Gallery of raw and symmetrized EM images of P14 WT CC transversal sections.** EM micrographs of WT P14 connecting cilia before and after symmetrization using CentrioleJ (see Methods), highlighting the presence of the CC inner scaffold (green arrowhead) and the Y-links (blue arrowhead). Scale bar: 200 nm. CC, connecting cilium; EM, electron microscopy; WT, wild type.
(TIFF)

**S4 Fig. Simulation of U-ExM signals from EM micrographs.** (a) Scheme representing the simulation pipeline as previously used in [16]. First, a raw EM micrograph was symmetrized using CentrioleJ (see Methods). Then, after contrast optimization, all the different structures (microtubules, inner scaffold, and Y-links) were segmented and submitted to a bandpass filter mimicking the limit of resolution obtained with classic fluorescence microscopy (200 nm). The simulated signals of each structure were then merged to reconstruct the final simulation. (b) Relative distance of each structure based on the reconstructed simulation. Top: representation of the line (white dashed) drawn to make the measurement of peak intensities for each structure (inner scaffold in green, microtubules in magenta, and Y-links in cyan). Bottom: peak intensity distances of each structure. (c) Merge of independent stainings of POC5 (green), CENTRIN (yellow), and CEP290 (cyan) to compare measurements with the simulation. The line drawn to make the measurement of peak intensities for each staining is represented with the white dashed line. Scale bar: 200 nm. (d) Peak intensity distances of each protein calculated from the merged image in (c). The data underlying all the graphs shown in the figure are included in the S1 Data file. EM, electron microscopy.
(TIFF)

**S5 Fig. CC POC5 and CEP290 growth over time.** Graphs representing the position (compared to MC distal end as the "0") and the length of the POC5 (a) and CEP290 (b) CC signals sorted by length. Each color depicts the age of the animals measured, confirming the timing of the growth of the 2 signals. The data underlying all the graphs shown in the figure are included in the S1 Data file. CC, connecting cilium.
(TIFF)

**S6 Fig. Impact of *Fam161a*$^{tm1b/tm1b}$ on RHODOPSIN and CONE OPSIN staining over time.** (a) Comparison of early rod outer segment development in WT and *Fam161a*$^{tm1b/tm1b}$ photoreceptors. Note that RHODOPSIN outlines the outer segment tubulin signal at P4 and then accumulates distally at P7. Scale bar: 500 nm. (b) Expanded P60 WT or *Fam161a*$^{tm1b/tm1b}$

retinas stained for CONE OPSIN (green) and tubulin (magenta). High magnification of the cone photoreceptors (right) shows that cone axonemes are also greatly impacted in *Fam161a*<sup>tm1b/tm1b</sup> retinas, revealing no obvious difference between cones and rods at P60 in mutant mice. Scale bar: low mag = 50 μm; High mag = 2 μm. WT, wild type.
(TIFF)

**S7 Fig. Inner scaffold protein stainings in the 2 mouse models of RP28.** (a) Expanded WT and *Fam161a*<sup>tm1b/tm1b</sup> photoreceptors stained for tubulin and FAM161A (top) or CENTRIN (bottom) at different ages. Note that P30/P60 represents a mixture of P30 or P60 images. Scale bar: 500 nm. White arrowheads point to centriole inner scaffold and green arrowheads point to CC inner scaffold, when present. (b) Expanded *Fam161a*<sup>GT/GT</sup> photoreceptors stained for tubulin and FAM161A (top) or POC5 (bottom) between P7 and P30. Scale bar: 500 nm. White arrowheads point to centriole inner scaffold and green arrowheads point to CC inner scaffold, when present. (c) Proportion of POC5-positive centrioles (left), FAM161A-positive centrioles (middle), or FAM161A-positive CC (right) in *Fam161a*<sup>tm1b/tm1b</sup> photoreceptors between P7 and P30. Note, for comparison, that the data for the percentage of POC5-positive CC are presented in Fig 4I. The data underlying all the graphs shown in the figure are included in the S1 Data file. CC, connecting cilium; WT, wild type.
(TIFF)

**S8 Fig. Impact of POC5 or FAM161A siRNA in human centrioles.** (a) Representative widefield images of expanded U2OS centrioles treated with siCtrl, siPOC5, or siFAM161A stained with tubulin (magenta) and FAM161A (gray) or POC5 (green). Inside each picture is depicted the total number of centrioles counted and the resulting averaged percentage from at least 3 independent experiments. Scale bar: 200 nm. (b) Mean fluorescence intensity of FAM161A under the indicated conditions. (c) Mean fluorescence intensity of POC5 under the indicated conditions. (d) Percentage of positive centrosomes (with FAM161A or POC5 staining) in siCtrl, siPOC5, or siFAM161A treated cells. (e) Representative widefield images of expanded U2OS centrioles treated with siCtrl or siFAM161A+ siPOC5 stained with tubulin (magenta) and inner scaffold protein (green). Scale bar: 200 nm; ≥3 independent experiments for each measurement. The data underlying all the graphs shown in the figure are included in the S1 Data file. siRNA, small interfering RNA.
(TIFF)

**S9 Fig. SPATA7 distribution in WT and *Fam161a*<sup>tm1b/tm1b</sup> photoreceptors.** Representative images of widefield (a) or confocal (b) expanded photoreceptors from 2-month-old WT or *Fam161a*<sup>tm1b/tm1b</sup> stained for SPATA7 (green) and tubulin (magenta). Scale bars: 500 nm. WT, wild type.
(TIFF)

**S10 Fig. EM gallery of P14 WT and P30 *Fam161a*<sup>tm1b/tm1b</sup> connecting cilium transversal sections.** EM micrographs of WT and mutant connecting cilia revealing the loss of MTD cohesion within the mutant axoneme. Note that Y-links are still observable in some MTDs, even in strongly affected CC where membrane invaginations are present within the axoneme (bottom right). Filled green arrowhead highlights the presence of the CC inner scaffold. Empty green arrowheads indicate the lack of CC inner scaffold. Blue arrowhead highlights the presence of Y-links. Magenta arrowheads show the membrane invaginations. Yellow arrowheads reveal opened B-microtubules. Scale bars: 200 nm. CC, connecting cilium; EM, electron microscopy; MTD, microtubule doublet; WT, wild type.
(TIFF)

**S11 Fig. Single particle average classification of WT or *Fam161a^tm1b/tm1b* microtubule doublets.** Representation of the classification (5 classes) of the particle averaging of microtubule doublets obtained from EM micrographs of P14 WT (a) or P30 *Fam161a^tm1b/tm1b* (b) (See Methods). The number of particles in each class is written in green, and the relative representation of each class is depicted below. For the most representative class of either WT or *Fam161a^tm1b/tm1b*, superimposition of the different structures (microtubules in magenta, Y-links in cyan, and the CC inner scaffold in green) was drawn for illustration purposes. Most representative classes were used for main figure. Scale bar: 20 nm. CC, connecting cilium; EM, electron microscopy; WT, wild type.
(TIFF)

**S12 Fig. RHODOPSIN mislocalization in late *Fam161a^tm1b/tm1b* retinas.** Expanded P60 WT or *Fam161a^tm1b/tm1b* retinas stained for RHODOPSIN (green) and tubulin (magenta). White arrows show RHODOPSIN signal at the level of photoreceptor cell bodies, notably around the nuclei in *Fam161a^tm1b/tm1b* retinas. Double-headed arrows reveal the difference of photoreceptor layer thickness between WT and *Fam161a^tm1b/tm1b*. Scale bar: 50 μm. WT, wild type.
(TIFF)

## Acknowledgments

We warmly thank Dror Sharon, Avigail Beryozkin, and Thomas Langmann for sharing their FAM161A-deficient mouse strains and Susanne Borgers, Sylvie Montessuit, and Jean-Claude Martinou for technical help as well as Maeva Le Guennec for help with S5 Fig. We thank the PFMU (UniGE) for EM samples preparation, the Bioimaging center (UniGE), and Marine. H. Laporte for providing the centrioles measurements in S1 Fig. We also thank Gabriel Aeschlimann (Ribosome Studio) for the design of the model in Fig 5.

## Author Contributions

**Conceptualization:** Paul Guichard, Virginie Hamel.

**Formal analysis:** Olivier Mercey, Eloïse Bertiaux, Alexia Giroud, Yashar Sadian.

**Funding acquisition:** Corinne Kostic, Yvan Arsenijevic, Paul Guichard, Virginie Hamel.

**Methodology:** Olivier Mercey, David C. A. Gaboriau, Ciaran G. Morrison.

**Resources:** Corinne Kostic, Ning Chang.

**Supervision:** Paul Guichard, Virginie Hamel.

**Validation:** Olivier Mercey.

**Visualization:** Olivier Mercey, Eloïse Bertiaux.

**Writing – original draft:** Olivier Mercey, Paul Guichard, Virginie Hamel.

**Writing – review & editing:** Olivier Mercey, Corinne Kostic, Ciaran G. Morrison, Yvan Arsenijevic, Paul Guichard, Virginie Hamel.

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
