## [Editor Report · Decision Letter 0]

25 Feb 2022

Dear Dr Hamel, 

Thank you for submitting the revision of your Review Commons manuscript entitled "The connecting cilium inner scaffold provides a structural foundation that protects against retinal degeneration" for consideration as a Research Article by PLOS Biology.

Your manuscript has now been evaluated by the PLOS Biology editorial staff as well as by an academic editor with relevant expertise and I am writing to let you know that we would like to send the revision back to the original reviewers. However, we need you first to complete your submission by providing the metadata that is required for full assessment. To this end, please login to Editorial Manager where you will find the paper in the 'Submissions Needing Revisions' folder on your homepage. Please click 'Revise Submission' from the Action Links and complete all additional questions in the submission questionnaire.

Once your full submission is complete, your paper will undergo a series of checks in preparation for peer review. Once your manuscript has passed the checks it will be sent out for review. To provide the metadata for your submission, please Login to Editorial Manager (https://www.editorialmanager.com/pbiology) within two working days, i.e. by Mar 01 2022 11:59PM. 

Given the disruptions resulting from the ongoing COVID-19 pandemic, please expect some delays in the editorial process. We apologise in advance for any inconvenience caused and will do our best to minimize impact as far as possible.

Kind regards,

Ines

--

Ines Alvarez-Garcia, PhD

Senior Editor

PLOS Biology

---

## [Decision Letter · Decision Letter 1]

5 Apr 2022

Dear Dr Hamel,

Thank you for submitting your revised Research Article entitled "The connecting cilium inner scaffold provides a structural foundation that protects against retinal degeneration" via Review Commons for publication in PLOS Biology. Thank you also for your patience as we completed our editorial process, and please accept my apologies for the delay in providing you with our decision. I have now obtained advice from the three original reviewers and have discussed their comments with the Academic Editor. 

Based on the reviews (attached below), we will probably accept this manuscript for publication, provided you satisfactorily address the remaining minor points raised by the reviewers. Please also make sure to address the data and other policy-related requests stated below my signature.

We expect to receive your revised manuscript within two weeks. 

*Published Peer Review History*

*Press*

Sincerely,

Ines

--

Ines Alvarez-Garcia, PhD,

Senior Editor

PLOS Biology

Fig. 1J; Fig. 2E-K; Fig. 3B, D, F, G; Fig. 4E-I, L; Fig. S1D, E, F; Fig. S2C, D; Fig. S4B, D; Fig. S5A, B; Fig. S7C and Fig. S8B-D

Please also ensure that figure legends in your manuscript include information on WHERE THE UNDERLYDING DATA CAN BE FOUND, and ensure your supplemental data file/s has a legend.

BLURB

Please also provide a blurb which (if accepted) will be included in our weekly and monthly Electronic Table of Contents, sent out to readers of PLOS Biology, and may be used to promote your article in social media. The blurb should be about 30-40 words long and is subject to editorial changes. It should, without exaggeration, entice people to read your manuscript. It should not be redundant with the title and should not contain acronyms or abbreviations. For examples, view our author guidelines: https://journals.plos.org/plosbiology/s/revising-your-manuscript#loc-blurb

Reviewers' comments

Rev. 1: Michael Housset and Michel Cayouette

The authors have addressed the major points that were raised in the initial review and their answers/additional data improve the manuscript. This is a very nice study with beautiful and convincing data that will be of great interest to the field. However, one of the authors' responses is debatable:

On page 17 of the revised manuscript, the authors corrected the reference of the Cep290 mutant to support their point of a potential independence between the Y-link structure and the CC-inner scaffold: "Consistently, Rachel and colleagues showed that Cep290ko/ko photoreceptors still possess the CC-inner scaffold, confirming the independence of these two structures".

In their paper, Rachel and colleagues show that connecting cilia fail to dock at the inner segment membrane in photoreceptors of Cep290ko/ko mice at P14 (Figure 4), but they also show that a 9+0 microtubule ring still assembles. As Rachel et al. did not stain for proteins specific of the CC-Inner segment (which was not known at the time) and considering the suboptimal resolution of their en-face EM sections, this data cannot be used to conclude about the presence or absence of a CC-inner scaffold in Cep290 KO mice. Additionally, in contrast to the Rachel et al study, nascent CC were shown to form in Cep290 KO photoreceptors at P10 by Potter and colleagues (2021). In this study, centrin was shown to decorate the nascent CC of Cep290ko/ko photoreceptors (Potter et al, 2021, Figure 6h).

Thus, the authors can only speculate on the interdependence of the Y-link and CC-inner segment scaffold as their presence has not been carefully investigated in Cep290 KO photoreceptors and should modify the text to that effect. 

Minor comment on the revised manuscript: 

Page 17 "While we found LCA5 mostly enriched at the bulge region, we also found some weak localization that might explain the reported interaction between LCA5 and FAM161A". Which localization are you referring to? The CC-inner scaffold? This is not clear.

Congratulations on this beautiful work.

Rev. 2:

The authors addressed all the points raised by this reviewer. The manuscript was satisfactory revised. This reviewer agrees with publication of this manuscript for PLOS Biology.

Rev. 3:

The authors have adequately addressed all of my previous comments submitted to Review Commons, both textually and experimentally, and have done a commendable and accurate job revising the manuscript accordingly. One minor comment: The titles of Table 1 and Table 2 are mentioned in the revised Supplementary material, but the tables are included in the main manuscript text. If this is supplementary information, they should be included there.

---

## [Editor Report · Decision Letter 2]

27 Apr 2022

Dear Dr Hamel,

On behalf of my colleagues and the Academic Editor, Renata Basto, I am happy to say that we can in principle accept your Research Article entitled "The connecting cilium inner scaffold provides a structural foundation that protects against retinal degeneration" for publication in PLOS Biology, provided you address any remaining formatting and reporting issues. These will be detailed in an email that will follow this letter and that you will usually receive within 2-3 business days, during which time no action is required from you. Please note that we will not be able to formally accept your manuscript and schedule it for publication until you have completed any requested changes.

PRESS

Sincerely, 

Ines

--

Ines Alvarez-Garcia, PhD 

Senior Editor 

PLOS Biology
